# The effect of different structural decoration geometries on vibration propagation in spider orb webs

G. Greco[1]*, V. F. Dal Poggetto[2], L. Lenzini[‡], F. Castellucci[3,4], N. M. Pugno[2,5]*

1 Department of Anatomy Physiology and Biochemistry Swedish University of Agricultural Sciences Uppsala, Uppsala, Sweden, 2 Laboratory for Bio-Inspired, Bionic, Nano, Meta, Materials & Mechanics, Department of Civil, Environmental and Mechanical Engineering, University of Trento, Trento, Italy, 3 Department of Biological, Geological and Environmental Sciences—University of Bologna, Bologna, Italy, 4 Zoology Section, Natural History Museum of Denmark—University of Copenhagen, Copenhagen, Denmark, 5 School of Engineering and Materials Science, Queen Mary University of London, London, United Kingdom

‡ Independent researcher.
☯ These authors contributed equally.
* gabriele.greco@slu.se (GG); nicola.pugno@unitn.it (NMP)

## Abstract

The stabilimentum, or structural decoration, in spider orb webs is a fascinating structure. While the species that construct stabilimenta and their building techniques are well-documented, the precise functions of these structures remain unclear. This knowledge gap arises from conflicting reports in the literature and the significant behavioral flexibility spiders exhibit when incorporating stabilimenta into their webs. Notably, spiders can build stabilimenta in various geometries, which may influence the dynamical properties of orb webs—a relationship that has yet to be quantitatively explored. In this study, we combined extensive field observations with computational simulations to address this gap. The fieldwork focused on documenting the range of stabilimentum geometries in *Argiope bruennichi*, while the simulations examined how these variations influence the propagation of elastic waves across the web. Our results suggest that the stabilimentum, acting as an additional inertial mass, does not significantly slow down the propagation of elastic waves generated by prey impact in the transverse and normal directions relative to the radial threads. However, when prey impact induces vibrations tangential to the spiral threads of the web, the presence of the stabilimentum enhances the spider's ability to detect prey location by allowing vibrations to reach a greater number of output points compared to webs without a stabilimentum. These findings deepen our understanding of the mechanical role of stabilimenta and provide new insights for the development of bio-inspired metamaterials, particularly those with tunable dynamic elastic properties.

**Data availability statement:** All the data necessary to understand the manuscript are contained in the supporting information.

**Funding:** G.G. was supported by the project "EPASS" under the HORIZON TMA Marie Skłodowska-Curie Actions Postdoctoral Fellowships - European Fellowships (project number 101103616). G.G., L.L. and F.C. were also supported by Aracnofilia – Italian Association of Arachnology. The funders had no role in study design, data collection and analysis, decision to publish, or preparation of the manuscript.

**Competing interests:** The authors have declared that no competing interests exist.

## Introduction

Spiders that construct orb webs use various types of silk to achieve high mechanical performance through an interplay of material properties and geometric design [1–3]. This synergy enables orb webs to effectively capture flying prey [4], facilitate prey detection [5,6], and minimize damage from impacts [1]. The web's adaptability is a response to environmental conditions and the spider's nutritional needs, showcasing the behavioral plasticity of web-building across different habitats [7–10].

Orb webs have long captivated human imagination, from inspiring philosophical reflection to influencing the development of emerging technologies [11–13]. A particularly intriguing aspect of these webs is the stabilimentum (plural: stabilimenta), a fascinating structural decoration of the web. In this paper, we use the term stabilimentum, commonly referred to as web decorations [14]. Stabilimenta are composed of thick layers of tough aciniform silk [15], arranged in distinct patterns within the web [16,17]. These silk decorations are typically produced using the spider's posterior median and lateral spinnerets [15], which weave silken bands in a zigzag fashion along radial threads [18]. Various species of spiders, belonging to different families, are capable of creating such ornamental structures, some of which may even incorporate non-silk elements [19–21]. The diversity and complexity of stabilimenta raise compelling questions about their potential functions and effects on the web's overall performance.

The functions of the stabilimentum remain a subject of ongoing debate [14,22]. Although the word suggests otherwise, it is well established that stabilimentum does not improve the structural stability of the web [23]. Conflicting data in the literature and the considerable geometric variability of these structures have made it challenging to pinpoint their exact role [23,24]. This complexity is heightened by the observation that, even within the same species, variations in stabilimentum geometry can fulfill different roles [25]. Several functions have been proposed, although it is likely that stabilimenta have multiple functions, including attracting flying insects while minimizing the attraction of potential predators [20,26–28], providing thermal protection [29], and collecting water [30]. For example, an hypothesis suggests that web decorations might attract insects by reflecting UV light [31]. This hypothesis is supported by recent studies showing that stabilimentum silk emits a stronger UV signal than other spider silks [32,33]. However, contrasting experiments on various araneid spiders report no evidence that stabilimenta exhibit higher UV reflectance than other silks [34,35]. Moreover, research on *Argiope* species shows that these spiders often omit stabilimenta from their webs, which suggests that producing these decorations involves a substantial cost and may even reduce prey capture efficiency [36,37]. Among the most widely supported hypothesis, one in particular states that stabilimenta act as a protective mechanism against predators, such as wasps or birds [28,38–41].

The presence of additional structural elements in a spider web is likely to influence its dynamic properties by modifying the web's geometry and, consequently, altering the pathways through which specific vibrations travel [42]. In particular, the propagation of elastic waves depends not only on the material properties but also

on the geometry of the structural components through which they move. Therefore, in principle, the propagation of elastic waves in webs containing a stabilimentum should differ from that in webs without such a structure. Thus, changes in the time required to perceive the prey impact on the web might affect predation. In this context, spiders are known to control how vibration signals emitted from trapped prey travel through the web via adjusting thread tension by adding elements [27,43,44]. However, no studies have yet investigated how web decorations affect vibration propagation in orb webs. Experimentally, this gap in knowledge arises from the considerable plasticity spiders exhibit when constructing web decorations. For instance, juvenile and adult spiders build stabilimenta with different geometries depending on their microhabitat, and these geometries can further vary in response to web damage or prey scarcity [41,45–48]. Understanding how stabilimentum geometry influences its functionality is therefore essential, yet no dedicated study has thoroughly explored this aspect.

Recent advancements in numerical simulations offer promising new tools for examining how different stabilimentum geometries affect the web's mechanical and vibrational properties, especially in terms of how prey impacts are detected and how vibrations propagate through the web [3,49]. These methods could significantly contribute to unravelling the stabilimentum's functions.

In this work, we surveyed three different populations of *Argiope bruennichi* (Scopoli, 1772) to quantify the occurrence and variability of stabilimentum geometries. Based on these geometries, we conducted numerical simulations to assess how elastic wave propagation properties are affected by the different stabilimenta. Our findings provide insights that could help clarify the stabilimentum's functional role and inspire the development of bio-inspired metamaterials with tunable dynamic elastic properties [12,13,42].

## Results and discussion

In this study we investigate the stabilimenta produced by three distinct populations of *Argiope bruennichi* (Figure S1) over three years between 2018–2020. We observed the stabilimentum produced by *Argiope bruennichi* with the posterior median and lateral pair of spinnerets (Fig 1a). The silken bands (if present) were woven in a zigzag fashion along radial threads (Figure 1b) [18].

In our survey, we identified several distinct stabilimentum geometries, which were categorized for the purposes of the subsequent simulation study (Fig 1c). Notably, juvenile specimens tend to construct a stabilimentum with a platform shape, roughly the size of the spider's leg span (platform type P). This platform is sometimes adorned with surrounding zigzag patterns (juvenile type J). In contrast, adult and larger female spiders typically create one of two stabilimentum types: the normal type N, which features two zigzag branches extending both upward and downward, and the reduced type R, which includes only the downward portion. In some instances, the stabilimentum is entirely absent (absent type A), while in others, it appears incomplete or thinly formed (drafted type D), with slender branches compared to the full stabilimentum.

As shown in Figure S2 and Table S1 in S1 File, the occurrence frequencies of these geometries are irregular, with stabilimenta absent in 50% of the cases. This variability aligns well with previous observations in the literature on the *Argiope* genus and other genera that produce web decorations [17,24,50,51].

The various stabilimentum geometries may also carry physical implications, particularly in terms of their effect on elastic wave propagation within the orb web. We hypothesized that these geometries could influence how vibrations, generated by prey impact, travel through the web. Building on recent research [3,49], we conducted numerical simulations to investigate the vibrations at the center of the web induced by prey producing vibrations at different positions and directions, using models based on the geometries shown in Fig 1c. For the sake of simplicity, our models did not include the web hub—the central region of the web, which is often removed after construction by certain species, such as *Argiope bruennichi*.

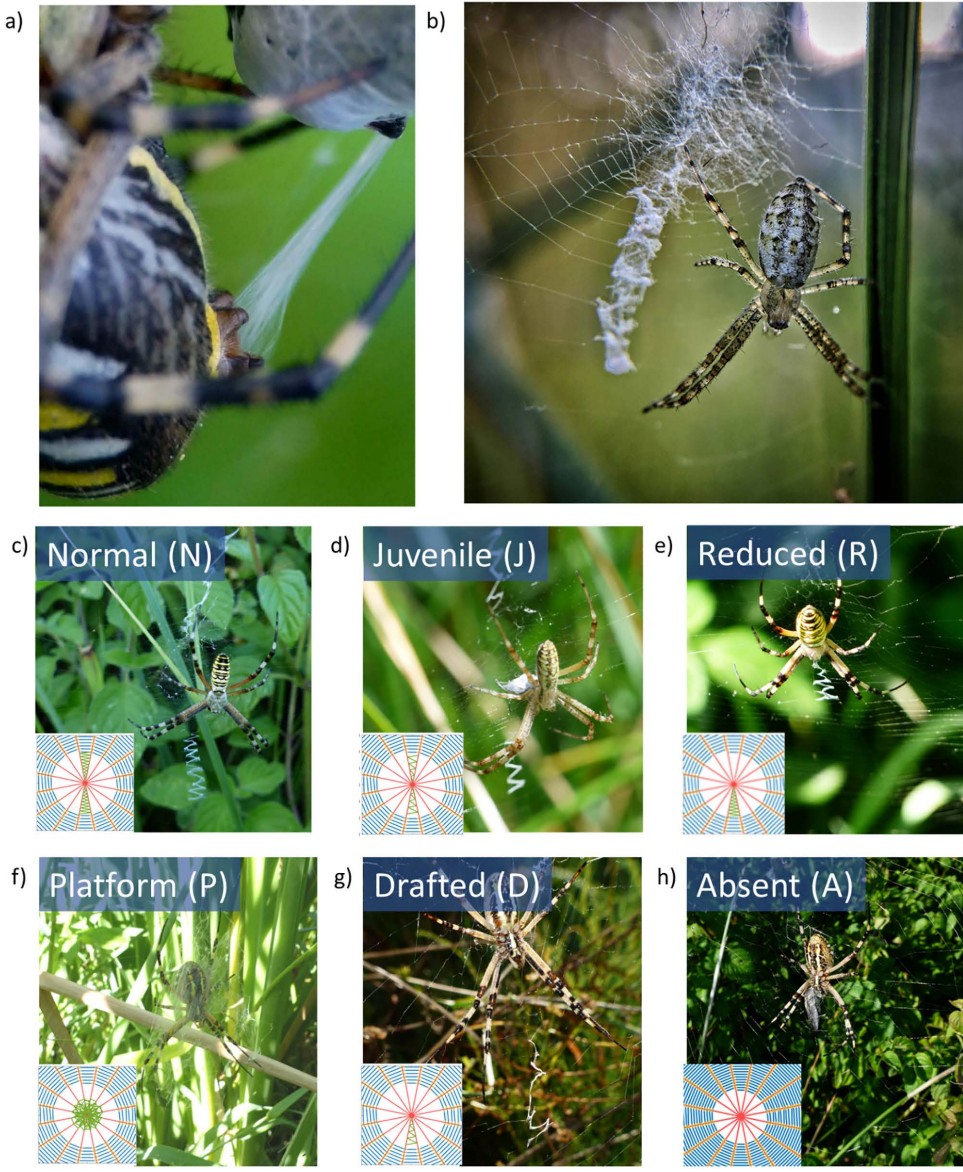

**Fig 1. The stabilimentum in *Argiope bruennichi*. a)** The production of aciniform silk by *A. bruennichi* when wrapping a prey. **b)** A juvenile A. bruennichi in the centre of its web with the stabilimentum (courtesy of Letizia Alleruzzo, Aracnofilia – Italian Association of Arachnology). **c– h)** The different types of stabilimentum observed in the webs: normal **(N)**, juvenile **(J)**, reduced **(R)**, platform **(P)**, drafted **(D)**, and absent (A, i.e., without stabilimentum).

To simulate the web's response, we created a computational model of a spider orb web using the finite element method [3]. The model incorporates 16 radial threads with pre-stressed elements (Fig 2a, see Materials and Methods and the S1 File supplementary section for details). The weight of the spider (20 mg) was evenly distributed, considering point masses, among the radial threads at the output points labelled as $O_1$-$O_{16}$. Prey impact was modelled by applying a sinusoidal half-period pulse (100 µN peak force and 5 ms duration), considering either (i) transverse vibrations (out-of-plane force, perpendicular to the plane of the orb web), (ii) tangential vibrations (with respect to the direction of spiral threads, in the plane of the orb web), or (iii) normal vibrations (with respect to spiral threads, in the plane of the orb web). Tangential

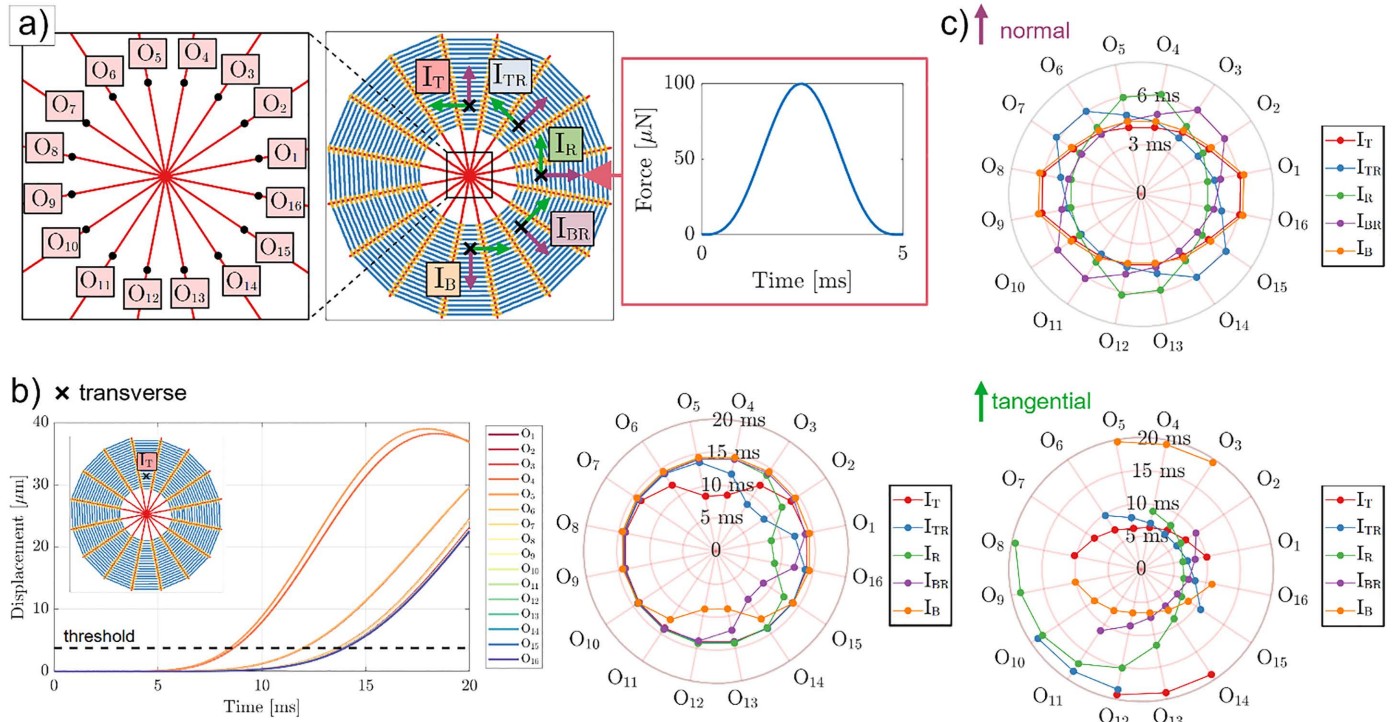

**Fig 2. Example of displacements and time to reach minimum displacement thresholds computed with the numerical model, considering an absent (A) stabilimentum. a)** A spider orb web is modelled consisting of 16 radial threads with output points O1-O16 corresponding to the points where the weight of the spider is distributed. Vibrations induced by the prey movement at various locations ($I_T$, $I_{TR}$, $I_R$, $I_{BR}$, and $I_B$) are simulated using the illustrated sinusoidal half-period pulse. The direction of the input force is either transverse to the plane of the orb web (black "x" marks), tangential to the spiral threads (green arrows), or normal to the spiral threads in the plane of the orb web (purple arrows). **b)** Results computed for the prey location $I_T$, shown for the [0,20] ms time window (left panel). The outputs points $O_4$ and $O_5$, corresponding to radial threads adjacent to the prey location, achieve a threshold displacement level (3.75 μm) before other radial threads. The results obtained for all impact points are summarized (right panel) considering the minimum time necessary for each output point to achieve a displacement levels of 3.75 μm. Angular coordinates indicate an output point associated with a radial thread and radial coordinates indicate a time measurement. **c)** Similar as **(b)**, but for input forces in the normal (top panel) and tangential directions (bottom panel).

vibrations are often called lateral in literature, but for the sake of clarity from the engineering point of view we will use the term tangential [23]. In addition, the directions of these forces are shown in Fig 2a. Five input points are considered to represent distinct sources of vibrations, labelled as $I_T$, $I_{TR}$, $I_R$, $I_{BR}$, and $I_B$, as also shown in Fig 2a.

The propagation speed of these input forces as well as the resulting displacement amplitudes at the output positions depend on several factors, which makes the comparison between these results non-trivial. In order to compare the results, we analyze, for each case, the time required for a given output point to achieve a certain displacement threshold level. These thresholds are computed, for each input direction (transverse, tangential, normal), by considering the five distinct input points ($I_T$ through $I_B$), averaging their maximum displacements, and considering the 10% of this average. Thus, we obtain the displacement thresholds of 3.75, 2.97, and 0.31 μm for the transverse, tangential, and normal input directions. The difference in these values is explained by the different stiffness of each thread with respect to the longitudinal and transverse directions, since the longitudinal stiffness is mostly owed to the material Young's modulus, while for the transverse direction, the stiffness is proportional to the pre-stress value.

Following this reasoning, Fig 2b (left panel) illustrates the displacements in a web without a stabilimentum for the $I_T$ impact point considering transverse prey vibration direction. The output points closest to the prey impact ($O_4$ and $O_5$)

display larger displacements more quickly than those farther from the impact site. Using the displacement threshold of 3.75 μm (black dashed line), we calculated the time required for each output point to reach this threshold. The results (Fig 2b, right panel) demonstrate that prey location significantly affects the time delay in a web without a stabilimentum. For instance, in Fig 2c, the output points $O_4$ and $O_5$, corresponding to radial threads adjacent to the prey impact, reached the displacement threshold the fastest (8.7 ms and 8.5 ms, respectively), followed by $O_3$ and $O_6$ (both at 12.0 ms). To verify the correlation between prey location and the time required for each radial thread to reach the displacement threshold, we performed similar calculations for prey impacts at points $I_{TR.}$ $I_R$, $I_{BR}$, and $I_B$ confirming this relationship (Fig 2b, right panel).

Similar analyses are performed for the cases of normal and tangential directions of prey vibration. In both these cases, the considered absolute displacements refer to the 2-norm for both x- and y-direction components of in-plane displacements (i.e., $u_i^2 = u_x^2 + u_y^2$, where $u_i$ are the absolute in-plane displacements, $u_x$ and $u_y$ are, respectively, the x- and y-direction computed displacements).

For the case of prey vibrations in the normal vibration direction (Fig 2c, top panel), the mean overall value to achieve the displacement threshold of 0.31 μm is 5 ms (circa), thus characterizing this prey vibration direction as the one with highest propagation speeds, which is explained mainly due to the considerable longitudinal propagation mode at radial threads, directly towards the orb web center. Also in this case, the location of prey vibrations is associated with the reduction in the time necessary to achieve a given displacement threshold for the pairs of radial threads adjacent to the prey location and also those opposing with respect to the web center. For instance, considering the $I_T$ location, the output points $O_4$ and $O_5$ present the minimum time of 4.11 ms, while the output points $O_{12}$ and $O_{13}$ present the minimum time 4.35 ms. This correlation is explained by the considerably larger longitudinal wave speed at radial threads, which cause vibrations to propagate across the orb web center with a much smaller time. Also, the reduction of these times is symmetric with respect to the prey vibration points, as expected due to the symmetry of the vibration direction.

For prey vibrations tangential to the spiral threads (Fig 2c, bottom panel), the times to achieve the minimum displacement threshold (2.97 μm) display a considerably more complex behavior with respect to the transverse vibrations case. Not all output points achieve the threshold level and this is shown, for instance, for the prey point $I_T$ (red points in Fig 2c, top panel), where only outputs $O_1$-$O_8$ and $O_{12}$-$O_{14}$ reach the displacement threshold. Particularly, for the output points $O_1$-$O_8$, the times to reach the displacement threshold range between 6.33 and 10.30 ms, thus revealing a somehow faster wave propagation than transverse vibrations. This could be explained by the fact that the vibration source elicits the corresponding spiral threads in the longitudinal direction and the radial threads in the normal one (longitudinal wave speeds are higher than transverse speeds, which cause a slight increase in the overall velocity). A similar behavior is observed for the other prey locations, with a special case being $I_R$, which indicates a larger quantity of output points achieving the displacement threshold. This behavior can be explained by the non-perfect symmetry of the orb web geometry.

The previous analysis of wave propagation induced by prey motion was then extended to investigate the effect of the various stabilimentum geometries shown in Fig 1c. For the inclusion of the stabilimentum geometry into the numerical model of the web, a residual stress value of 0.1 Pa was applied to the stabilimentum elements to ensure numerical stability when reaching an equilibrium state of stress in the simulated orb webs. To enable comparison, the minimum time required for each web to achieve the displacement threshold was calculated for each prey vibration direction and compared to the baseline case, where no stabilimentum was present (case A in Fig 1c).

The first considered case is that of vibrations in the transverse direction with respect to the orb web plane. We present, in Fig 3, the times necessary for the transverse displacements at the output points to reach the set displacement threshold (3.75 μm, in this case). For this situation, the introduction of stabilimentum produces a negligible delay (in the order of μs) in comparison to the overall times of propagation (in the order of ms, see S1 File Supplementary information for detailed discussion). The presence of a stabilimentum introduces greater delays when the prey is located in the same direction as the stabilimentum structure. We recall that the transverse waves propagating in both radial and spiral threads have a speed proportional to the value of pre-stress in these structural elements, which, for the case of transverse prey

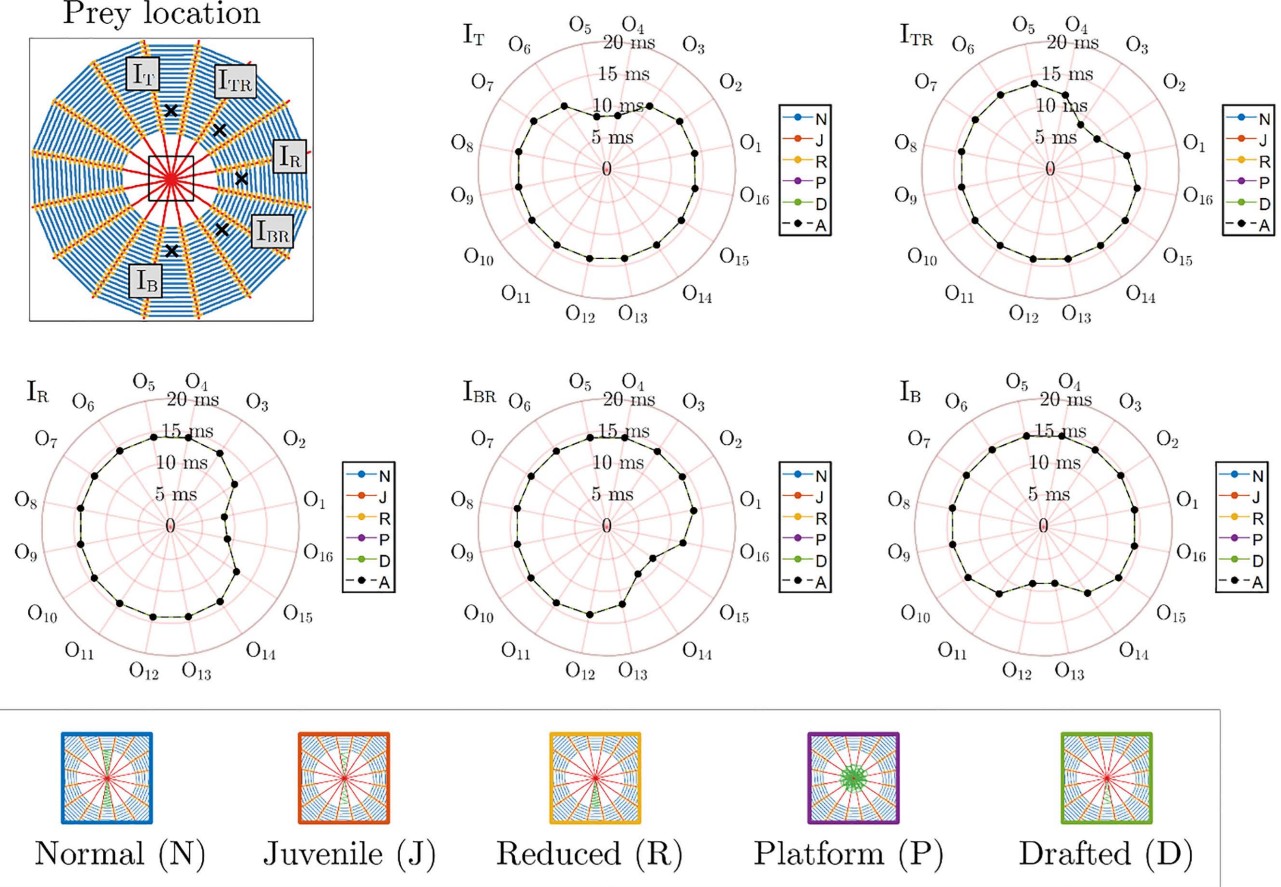

**Fig 3. Effects of the presence of stabilimentum concerning prey vibration acting in the direction transverse to the orb web plane.** Time intervals (in ms) necessary for the output points located at each radial thread ($O_1$-$O_{16}$) of orb webs with stabilimentum (N, J, R, P, D, represented in blue, red, yellow, purple, and green continuous lines, respectively) to achieve the minimum displacement threshold (3.75 µm) when compared to the baseline case (absent stabilimentum, A, black dashed lines). The considered prey locations ($I_T$, $I_{TR}$, $I_R$, $I_{BR}$, and $I_B$) are shown for reference, each marked with a black "x" (top left panel). The presence of stabilimentum does not influence significantly the time necessary for the output points between the web centre and the prey to achieve the displacement threshold. Indeed, the colored lines seem overlapped.

vibrations, are the only type of elicited motion (i.e., there is no interaction between longitudinal and transverse waves in the web). As a consequence, it is reasonable to assume that the observed delay for each case is due to the added inertia of the stabilimentum on the orb web, i.e., increasing the local mass of the structure and hindering its displacements. It is important to note, however, that all the time delays associated with the introduction of stabilimentum are very small when compared to the overall traveling times (for comparative purposes, consider 20 µs/10 ms = 0.2%), and are not likely to hinder prey detection.

We performed the same analysis concerning prey vibration in the direction normal to the spiral threads. Fig 4a reports a comparison between the displacements computed at threads $O_4$, $O_5$, $O_{12}$, and $O_{13}$, for the cases without stabilimentum (absent, A), and with stabilimentum (normal, N), for the cases $I_T$ and $I_R$ (left and right panels, respectively). We notice that the inclusion of stabilimentum of the N type has a mild effect considering both the amplitude and time necessary to achieve the displacement threshold levels (0.31 µm, in this case) for the prey location point aligned with the stabilimentum ($I_T$). On the contrary, although the inclusion of stabilimentum of the N type decreases the overall amplitude for the case

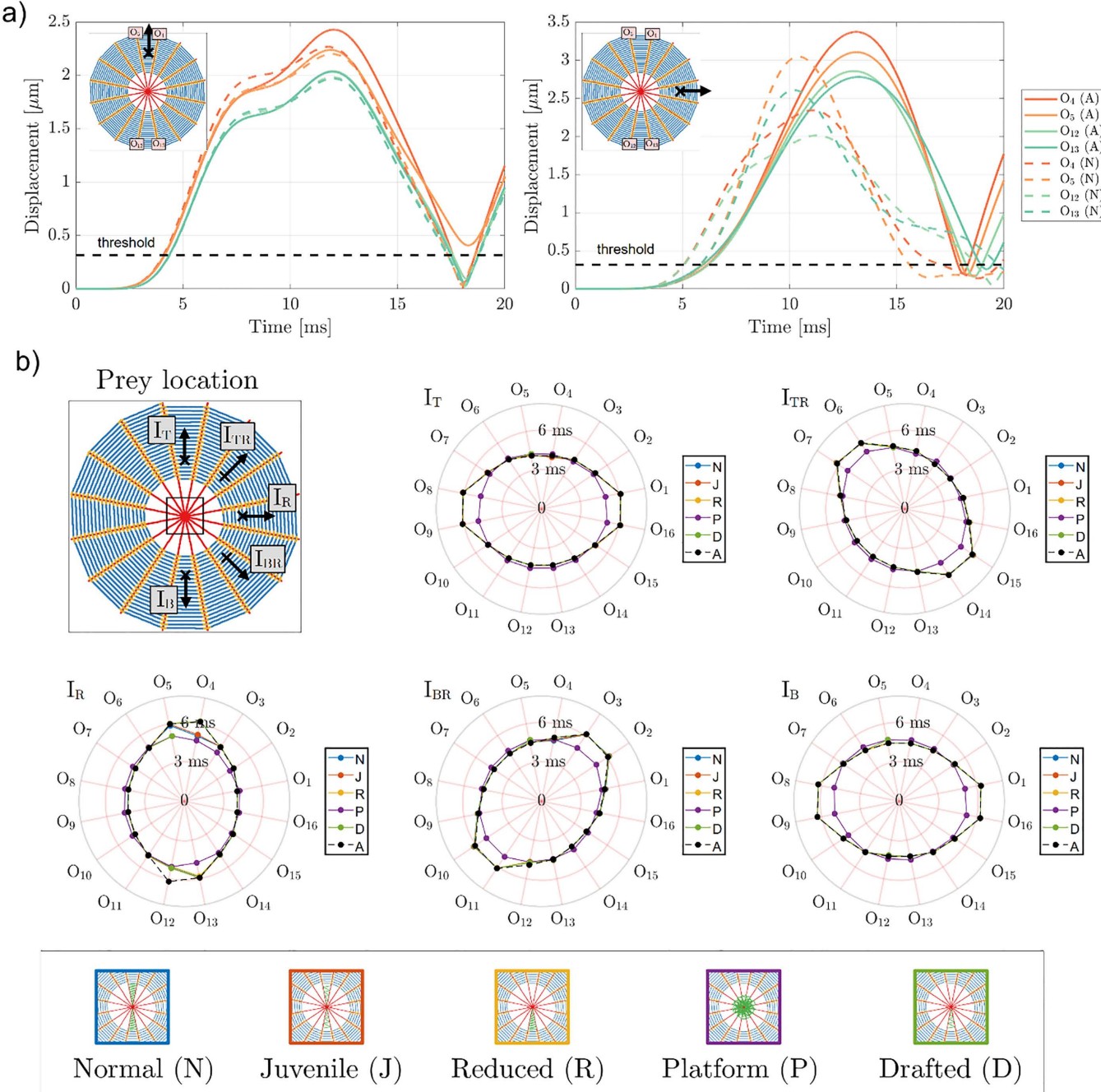

**Fig 4. Effects of the presence of stabilimentum concerning prey vibration acting in the direction normal to the spiral threads. a)** Displacements at outputs O4 (dark orange), O5 (light orange), O12 (light green), and O13 (dark green) for the cases without stabilimentum (absent, A, continuous colored lines) and with stabilimentum (normal case, N, dashed colored lines). Black dashed lines indicate the displacement threshold (0.31 μm). The prey location and vibration direction are marked using a black "x" and an arrow, respectively. **b)** Time intervals (in ms) necessary for the output points located at each radial thread ($O_1$-$O_{16}$) of orb webs with stabilimentum (N, J, R, P, D, represented in blue, red, yellow, purple, and green continuous lines, respectively) to achieve the minimum displacement threshold when compared to the baseline case (absent stabilimentum, A, black dashed lines). The considered prey locations ($I_T$, $I_{TR}$, $I_R$, $I_{BR}$, and $I_B$) are shown for reference (top left panel). In general, the presence of stabilimentum does not influence significantly the time necessary for the output points between the web center and the prey to achieve the displacement threshold (notice the overlap of the coloured lines). An exception to this is the platform stabilimentum (P), which presents an almost constant response (3 ms), regardless of the prey location.

where the prey location is orthogonal to the direction of the stabilimentum ($I_R$), it also decreases the necessary time to reach the threshold.

Furthermore, Fig 4b reports the necessary time intervals to achieve the displacement threshold for each case of prey location ($I_T$, $I_{TR}$, $I_R$, $I_{BR}$, $I_B$, depicted at the top left panel) and stabilimentum type (N, J, R, P, D, depicted at the bottom panel). The absent case is also given, for reference (same as Fig 2c, top panel). We notice that for almost all cases ($I_T$, $I_{TR}$, $I_{BR}$, and $I_B$) there is no significant difference between the cases with and without stabilimentum (the lines overlap in Fig 4b), with the exception of the platform case (P). In this case, the stabilimentum seems to produce a nearly constant and direction-independent time to achieve the threshold. Although this time is shorter than some other cases, this may, in fact, be detrimental for the input localization at the center of the web, since it may hinder a directional cue for detection. Interestingly, there is a subtle decrease in the response times for the case where the prey location is orthogonal to the direction of stabilimentum ($I_R$). This also appears as a secondary effect of the existence of stabilimentum, given the modest reduction in time necessary to achieve considerable displacement levels at the direction of the prey vibrations. These subtle differences may also be explained due to the fact that the orb web does not present perfect symmetry, which can also be verified in Fig 2c (bottom panel).

Finally, we performed the same analysis considering the tangential direction of prey vibration (as depicted in Fig 2a). In Fig 5a, we report a comparison between the displacements computed at threads $O_4$, $O_5$, $O_{12}$, and $O_{13}$, for the cases without stabilimentum (absent, A), and with stabilimentum (normal, N). We show, for illustrative purposes, the results computed for the cases $I_T$ and $I_R$ (Fig 5a, left and right panels, respectively). We notice that, for tangential vibration, the inclusion of stabilimentum has a significant effect on (i) the amplitude of waves reaching the output points and (ii) an overall reduction in the time necessary to achieve the displacement threshold level (2.97 µm, in this case). Furthermore, for the tangential case, the variation in the displacement amplitudes at the output points does not follow the same tendency for all threads (i.e., these might increase or decrease), which can be explained by the asymmetrical application of force on the orb web.

Next, Fig 5b summarizes the necessary time intervals to achieve the displacement threshold for each case of prey location ($I_T$, $I_{TR}$, $I_R$, $I_{BR}$, $I_B$, depicted at the top left panel) and stabilimentum type (N, J, R, P, D, depicted at the bottom panel). The absent case is also given, for reference (same as Fig 2c, bottom panel). Two fundamental observations can be made concerning these results.

The first one is that the inclusion of the stabilimentum allows some outputs that previously did not reach the displacement threshold level to now achieve it. These output points are mostly located opposed to the prey, with respect to the orb web center, as it can be noticed in the case of stabilimentum of types N and J, for the prey location point $I_T$ (Fig 5b, top middle panel). For the same case, however, this improvement was not observed when the stabilimentum was not located between the prey and the orb web center (as in cases R and D, which are located opposed to $I_T$). To reinforce this interpretation, we note that for the prey location $I_B$, almost all stabilimenta present an improvement in sensitivity, with the exception of the D case, which, due to its nearly circular distribution, seems to be less beneficial with respect to highly direction stabilimenta, but yields an improvement for all directions (also $I_{TR}$, $I_R$, $I_{BR}$). The second observation is that the actual time interval necessary to achieve the threshold is just marginally reduced with respect to the case without stabilimentum (absent, A), especially considering the regions between the orb web center and the prey, which were already able to sense the prey, in any case.

Thus, we conclude that for tangential vibrations, the slight reduction in prey detection time is largely offset by a significant increase in sensitivity, as more threads become involved in sensing the prey. This effect can be attributed to the enhanced connectivity of radial threads near the center of the orb web, which likely facilitates wave propagation within the web.

In principle, changes in the time required to perceive prey impact could influence the spider's response, ultimately affecting predation. Our results indicate that (i) transverse vibrations exhibit a negligible delay, while (ii) normal and

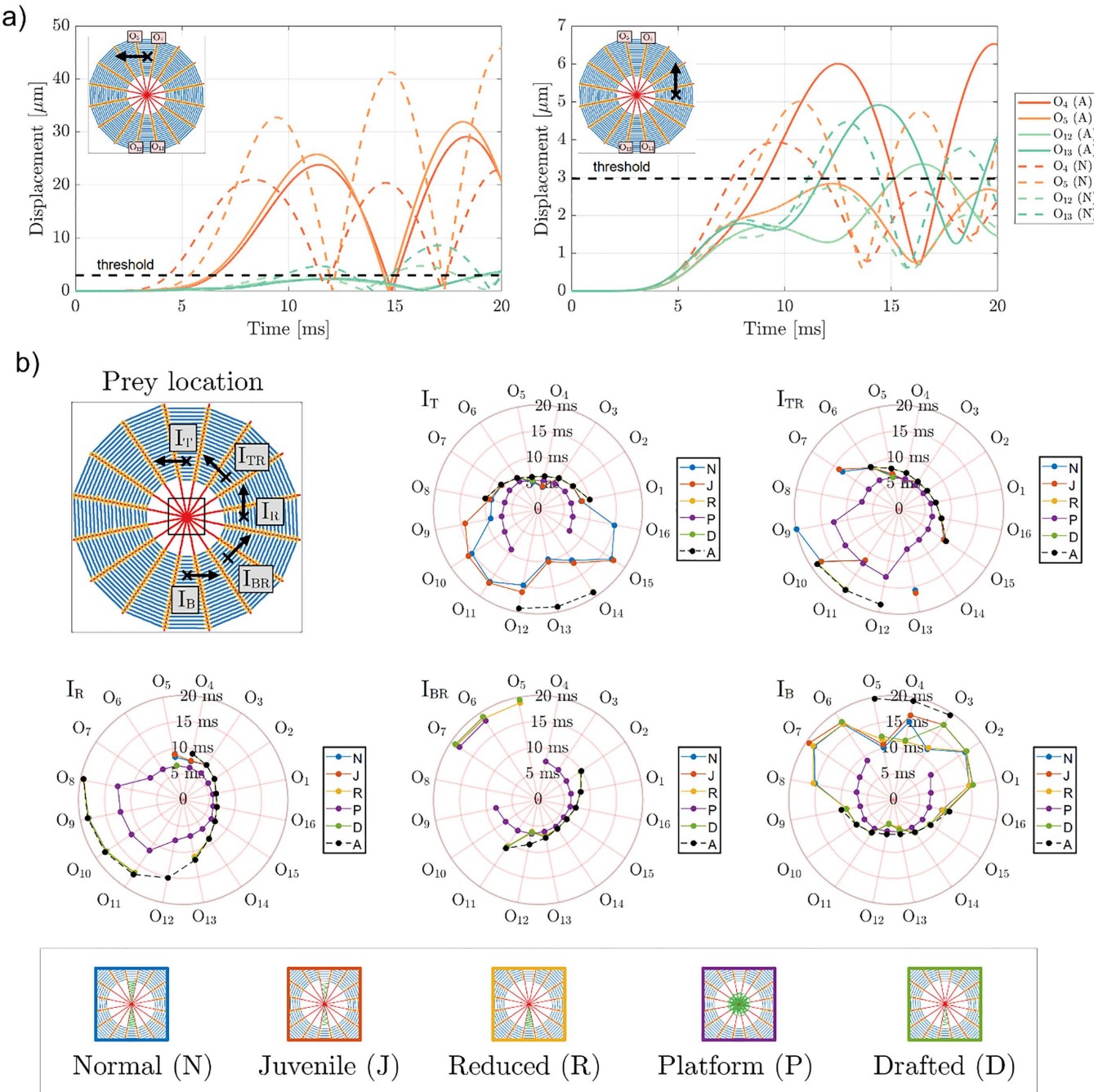

**Fig 5. Effects of the presence of stabilimentum concerning prey vibration acting in the direction tangent to the spiral threads. a)** Displacements at outputs O4 (dark orange), O5 (light orange), O12 (light green), and O13 (dark green) for the cases without stabilimentum (absent, A, continuous colored lines) and with stabilimentum (normal case, N, dashed colored lines). Black dashed lines indicate the displacement threshold (2.97 μm). The prey location and vibration direction are marked using a black "x" mark and an arrow, respectively. **b)** Time intervals (in ms) necessary for the output points located at each radial thread ($O_1$-$O_{16}$) of orb webs with stabilimentum (N, J, R, P, D, represented in blue, red, yellow, purple, and green continuous lines, respectively) to achieve the minimum displacement threshold when compared to the baseline case (absent stabilimentum, A, black dashed lines). The considered prey locations ($I_T$, $I_{TR}$, $I_R$, $I_{BR}$, and $I_B$) are shown for reference (top left panel). In general, the presence of stabilimentum is associated with a marginal decrease of time necessary for the output points between the web center and the prey to achieve the displacement threshold but allow the opposing threads to achieve the threshold levels, whenever stabilimentum is present between the prey and the web center. When looking at the figure, beware of the overlap of some lines (e.g., A and D cases for the output O8-O11).

tangential vibrations show only a minor reduction in the time needed to reach the displacement threshold. The most pronounced reductions in response time occur in the platform configuration, which may, however, compromise directional cues essential for prey localization. Interestingly, the presence of the stabilimentum appears to enhance the sensitivity of the orb web center to tangential vibrations, preserving vibrational cues that aid in localization. In this context, our findings align with those of Watanabe and Gong et al., who demonstrated that web geometry plays a crucial role in shaping a spider's reaction time and predation behavior [47,48].

The argument that stabilimenta could affect prey perception should be done considering typical spider response latencies—usually around 30 ms [52,53]. A delay in perception measured in microseconds is unlikely to significantly impact the spider's reaction time, but for the vibrations tangentially directed the stabilimentum could play an important role in prey perception due to the increase in vibrations amplitude. Moreover, the spider's position on the web could play a role in mitigating/enhancing such change in prey perception time, since different positions alter spider weight distribution on the web [54]. For instance, *Argiope bruennichi* often adopts an X-like leg position when stabilimentum geometries N, J, R, and D are present, potentially reducing the time to detect prey impacting in the $I_{TR}$ and $I_{BR}$ cases. Additionally, the spider's position may serve biological functions—such as mimicking the geometry of the stabilimentum to evade detection by potential predators—which, however, fall outside the scope of this study.

Another important aspect to consider when discussing these results is the fact that also the movements of the spider can affect the dynamic properties of the web. For example, a recent research suggests that stabilimentum structures may work in concert with the spider's abdominal movements to achieve maximum defensive efficiency [55].

In conclusion, it is possible that stabilimenta present a trade-off between prey detection and predator avoidance [28,56], though this relationship requires further investigation. Our results contribute to a better understanding of the role stabilimenta play in shaping the dynamic properties of orb webs, revealing that prey detection seems to be influenced only by vibrations propagating tangentially to the spiral threads. However, from a biological standpoint, the high variability in stabilimentum geometry suggests that the observed differences in elastic wave propagation are unlikely to have a consistent or significant functional role. For this reason, our findings are better interpreted from an engineering perspective. They open up new possibilities for the design of bio-inspired metamaterials with tunable anisotropic wave propagation properties—for example, spiderweb-inspired lattices that allow controlled delays in wave transmission along specific pathways.

## Conclusion

The stabilimentum is a structure produced by spiders to decorate their orb webs, yet its functions remain largely unknown, with many hypotheses still untested or lacking experimental validation. In this study, we conducted a field survey across three distinct *Argiope bruennichi* populations to assess the diversity of stabilimentum geometries. Based on these observations, we performed numerical simulations on simplified web geometries to investigate how these decorations influence vibration propagation within the orb web.

Our survey revealed substantial variability, with no consistent geometric pattern in stabilimentum construction and an absence of decoration in 50% of the observed webs. Simulations showed that stabilimenta induce negligible delays in prey perception for transverse vibrations and only minor delays for normal and tangential vibrations, primarily due to the added inertial mass. However, for tangential vibrations, the presence of stabilimenta may enhance the spider's ability to localize prey, likely due to increased connectivity at the center of the orb web.

We hope this study contributes to ongoing discussions within the scientific community by providing new quantitative evidence while also offering insights for the design of bio-inspired metamaterials.

## Materials and methods

### Field survey

We investigated the stabilimenta produced by three distinct Sardinia (Italy) populations of *Argiope bruennichi* [57] (Figure S1 in S1 File) over three years between 2018–2020. The stabilimenta were classified according to Fig 1c, and

documented with pictures. Since to documents the stabilimenta type we only took pictures without taking the animals, no permission was needed. We acknowledge that additional geometrical variations may occur in nature; however, for the sake of simplicity, we restricted our simulations to the geometries depicted here.

### Numerical simulations

A computational model of a spider orb web consisting of 16 radial threads with pre-stressed elements (circa 50 and 10 MPa mean levels of initial pre-stress for the radial and spiral threads, respectively) was implemented using the finite element method (see S1 File Supplementary information for details). For the current simulations, a spider with a mass of 20 mg is considered, whose weight is evenly distributed between all the radial threads at the points labelled as $O_1$-$O_{16}$, distant 10 mm from the geometrical center of a 300-mm radius orb web. Then, a sinusoidal half-period pulse with amplitude 100 µN and width 5 ms (i.e., 100 Hz frequency), representing (i) an out-of-plane force (transverse to the orb web plane), (ii) in-plane normal, and (iii) in-plane tangential forces exerted by a moving prey, is applied to the points labelled as $I_T$, $I_{TR}$, $I_R$, $I_{BR}$, and $I_B$, respectively, at varying angles (see Fig 2a) and distant 150 mm from the web center. The displacement levels at the considered output points ($O_1$-$O_{16}$) are then computed in a time transient simulation (see S1 File Supplementary information for details). Fig 2a shows the labels of the output points, a representation of the modelled orb web, various prey location points, and the considered input signal. These simulations should be interpreted qualitatively rather than quantitatively, as several parameters may influence their outcome.

## Supporting Information

**S1 File. Supplementary information: file with all the supplementary information related to this manuscript, including the simulation details.**
(DOCX)

## Acknowledgments

G.G., L.L. and F.C. would like to thank Aracnofilia – Italian Association of Arachnology. We thank Massimo De Agrò for suggesting relevant literature.

## Author contributions

**Conceptualization:** G. Greco, V. F. Dal Poggetto, N. M. Pugno.

**Data curation:** G. Greco, V. F. Dal Poggetto, L. Lenzini.

**Formal analysis:** V. F. Dal Poggetto, L. Lenzini.

**Investigation:** V. F. Dal Poggetto, L. Lenzini.

**Methodology:** G. Greco, V. F. Dal Poggetto, L. Lenzini.

**Supervision:** G. Greco, N. M. Pugno.

**Validation:** G. Greco.

**Visualization:** G. Greco, F. Castellucci.

**Writing – original draft:** G. Greco, V. F. Dal Poggetto.

**Writing – review & editing:** G. Greco, V. F. Dal Poggetto, F. Castellucci, N. M. Pugno.

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
