## [Decision Letter · Decision Letter 0]

9 Dec 2024

PONE-D-24-51796The effect of different stabilimentum geometries on vibration propagation in orb websPLOS ONE

Dear Dr. Greco,

Thank you for submitting your manuscript to PLOS ONE. After careful consideration, we feel that it has merit but does not fully meet PLOS ONE’s publication criteria as it currently stands. Therefore, we invite you to submit a revised version of the manuscript that addresses the points raised during the review process.

Both reviewers questioned: 1) the fact that the study only analysed transverse vibration without including the information on longitudinal vibration; and following that, 2) the authors’ discussion on the function of the stabilimenta. I am afraid their concerns are fundamental to the significance of the study. In order to reconsider this study for publication, we would require a considerable amount of revision, restructuring the analysis to include longitudinal vibration. Both reviewers included a number of useful suggestions, so I hope the authors make use of them to improve the manuscript.

We look forward to receiving your revised manuscript.

Kind regards,

Shoko Sugasawa

Academic Editor

PLOS ONE

Journal Requirements:

4. Thank you for stating the following financial disclosure: [G.G. is supported by the project “EPASS” under the HORIZON TMA MSCA Postdoctoral Fellowships - European Fellowships (project number 101103616).]. Please state what role the funders took in the study. If the funders had no role, please state: "The funders had no role in study design, data collection and analysis, decision to publish, or preparation of the manuscript." If this statement is not correct you must amend it as needed. Please include this amended Role of Funder statement in your cover letter; we will change the online submission form on your behalf.

5. Please expand the acronym “MSCA” (as indicated in your financial disclosure) so that it states the name of your funders in full. This information should be included in your cover letter; we will change the online submission form on your behalf.

6. Thank you for stating the following in the Acknowledgments Section of your manuscript: [G.G. is supported by the project “EPASS” under the HORIZON TMA MSCA Postdoctoral Fellowships - European Fellowships (project number 101103616). G.G., L.L. and F.C. were also supported by Aracnofilia – Italian Association of Arachnology. We thanks Massimo De Agrò for suggesting relevant literature.  ] We note that you have provided funding information that is not currently declared in your Funding Statement. However, funding information should not appear in the Acknowledgments section or other areas of your manuscript. We will only publish funding information present in the Funding Statement section of the online submission form. Please remove any funding-related text from the manuscript and let us know how you would like to update your Funding Statement. Currently, your Funding Statement reads as follows: [G.G. is supported by the project “EPASS” under the HORIZON TMA MSCA Postdoctoral Fellowships - European Fellowships (project number 101103616).] Please include your amended statements within your cover letter; we will change the online submission form on your behalf.

7. Please remove your figures from within your manuscript file, leaving only the individual TIFF/EPS image files, uploaded separately. These will be automatically included in the reviewers’ PDF**.**

8. Please include captions for your Supporting Information files at the end of your manuscript, and update any in-text citations to match accordingly. Please see our Supporting Information guidelines for more information: http://journals.plos.org/plosone/s/supporting-information .

Reviewers' comments:

Reviewer's Responses to Questions

**Comments to the Author**

1. Is the manuscript technically sound, and do the data support the conclusions?

Reviewer #1: No

Reviewer #2: Yes

2. Has the statistical analysis been performed appropriately and rigorously? 

Reviewer #1: N/A

Reviewer #2: I Don't Know

3. Have the authors made all data underlying the findings in their manuscript fully available?

Reviewer #1: Yes

Reviewer #2: Yes

4. Is the manuscript presented in an intelligible fashion and written in standard English?

Reviewer #1: Yes

Reviewer #2: Yes

5. Review Comments to the Author

Reviewer #1: Orb webs of diurnal spiders often have silk attachments over or around web hub, the function of which are the subject of long-lasting controversy. This study focused the question, paying attention for the variation of form of silk decoration. The study simulated the vibrations emitted by prey caught in webs and examined the effect of different silk decoration morphologies on their propagation velocity.

The questions addressed in this study are worth investigating. The author's hypothesis that silk decoration may influence prey detection is interesting. However, I have concerns about the authors’ conclusions.

Simulation results showed that vibration transmission was more delayed in the web with silk decoration. I agree with that these results were opposite to the expectation of prey detection enhance hypothesis. Nevertheless, spiders may use amplitude information to detection threshold which was not examined in this study. Moreover, the authors did not examined longitudinal vibration which may also be used by spiders. Therefore, it is premature to conclude that silk decoration does not link to enhancing prey capture efficiency from the results of this study.

Standing on this, I cannot agree with the authors’ another conclusion about defensive function. There are various hypothesis for the function of silk decoration, the lack of supporting evidence for prey capture/detection hypothesis does not always indicate the support for defence hypothesis.

I recommend that the authors conduct additional analysis or that the manuscript be reorganized so that the authors only state conclusions based on their findings.

I have other concerns. The authors use the term “stabilimentum” throughout the manuscript. However, the term comes from “to stabilize web”. Now, a lot of other functions were proposed, as mentioned in introduction, I suppose that“silk decoration” is a more appropriate term.

In introduction, the rationale of the connection between the presence of silk decoration and vibration propagation was not fully developed. No previous studies about silk decoration is directly connected with vibration. Therefore, I recommend the authors to explain more throughly why the effect of silk decoration on vibration propagation should be examined and what outcomes would be expected in introduction.

L77. I happened to be aware that the phrase “is added to neighboring radii in a zigzag manner” coincides perfectly with one appeared in Foelix’s book. Although the book is referred, I recommend the authors to rephrase it.

Figure 1c. The scheme for Juvenile type is somewhat different from actual silk decoration appeared in the photo.

Reviewer #2: I am a biologist, and have no expertise in bio-materials, and I thus cannot evaluate the major aspects of this paper relating to quantification of elastic wave propagation and the resulting perspectives on bio-inspired metamaterials. I have only two comments in this area. I wonder why the authors used only out-of-plane vibrations, when previous work (Klärner, D., and F. G. Barth. 1982. Vibratory signals and prey capture in orb-weaving spiders (Zygiella x-notata, Nephila clavipes; Araneidae). Journal of Comparative Physiology A: Neuroethology, Sensory, Neural, and Behavioral Physiology 148:445–455) concluded that, for the spider, the most important vibrations of non-sticky lines in orbs are longitudinal rather than tranverse. This gives reason to question the biological implications of this work on elastic wave propagation, but does not necessarily impact implications regarding bio-inspired metamaterials. A second comment is that the spider’s own weight, while she rests at the hub, must alter the tensions on different radii and hub lines substantially. Especially strong increases in tensions would be expected at the two sites held by the tarsi of the spider’s legs IV (increasing tensions above and decreasing just below these tarsi); such tension modifications would be expected to affect transmission of vibrations. I do not see how this effect was included in the simulations.

On the biological side of the paper, I believe that the findings in this study are of doubtful importance for discussion of the possible functions of stabilimenta for the spiders. I would advise the authors to omit most of what they say about the functions of stabilimenta and focus their paper on elastic wave propagation. This would require substantial, but only a relatively simple modifications of the ms.

My doubts regarding the function(s) of stabilimenta stem from the following points. 1) The wide variation that the authors document in stabilimentum design (long known in this and other species) is itself an argument against the importance of its effects on elastic wave propagation; if more rapid propagation were important in promoting prey capture, then one would not expect webs with more elaborate stabilimenta (and vice versa) (the variation in stabilimentum design is explained by alternative hypotheses for stabilimentum function such as protection from visually orienting predators). 2) The fact that many orb weaving species remove the entire central portion of the hub would require additional explanation if propagation of elastic waves in the hub has an important effect promoting prey capture. 3) The first vibrations conveying the information needed by the spider at the hub regarding the location where the prey is located do not arrive from the central portions of the hub, but from beyond the hub. Propagation of elastic waves through the hub, which in Figs. 2a and 2c,d seem to suggest was the phenomenon that was modeled in this study, is not obviously related to this. 4) The constrained positions of the spider’s legs, whose mimicry of the form of the stabilimentum in many other species (including other Argiope species) is explained by alternative hypotheses of protection from predators and prey capture, but is not explained by the elastic wave propagation hypothesis.

Minor points:

As the authors say, their conclusion from their survey that stabilimentum forms vary intra-specifically is not new. As such, it is data perhaps appropiate for a specialist spider journal, not PLoS. Perhaps they can document contrasts between their observations and previous studies.

Line 36 The claim that orbs “optimize mechanical performance” in biological terms (costs of materials, construction behavior and maintenance versus payoff in materials and energy from prey captured due to vibration transmission, interception and retention of prey in nature) is not documented in the references cited (nor in any analysis of any orb web, despite frequent claims to the contrary in the literature). Various factors have been studied separately, as mentioned in the following lines in the ms; but the balance is unknown (and extremely difficult to document convincingly). The claim for optimization needs to be removed.

6. PLOS authors have the option to publish the peer review history of their article (what does this mean? ). If published, this will include your full peer review and any attached files.

**Do you want your identity to be public for this peer review?** For information about this choice, including consent withdrawal, please see our Privacy Policy .

Reviewer #1: No

Reviewer #2: No

---

## [Author Response · Author response to Decision Letter 1]

20 Jun 2025

Reviewer 1:

Orb webs of diurnal spiders often have silk attachments over or around web hub, the function of which are the subject of long-lasting controversy. This study focused the question, paying attention for the variation of form of silk decoration. The study simulated the vibrations emitted by prey caught in webs and examined the effect of different silk decoration morphologies on their propagation velocity.

The questions addressed in this study are worth investigating. The author's hypothesis that silk decoration may influence prey detection is interesting. However, I have concerns about the authors’ conclusions.

Reply:

We thank the reviewer for this valuable comment. In response, we have significantly improved the manuscript, and we hope that it is now considered suitable for publication. Notably, we have added new data and additional simulations (the new data and text were not highlighted for the sake of readability), which have led to a substantial revision of the key message of the work. We are grateful to the reviewer for providing important suggestions that contributed to this major improvement.

Simulation results showed that vibration transmission was more delayed in the web with silk decoration. I agree with that these results were opposite to the expectation of prey detection enhance hypothesis. Nevertheless, spiders may use amplitude information to detection threshold which was not examined in this study. Moreover, the authors did not examined longitudinal vibration which may also be used by spiders. Therefore, it is premature to conclude that silk decoration does not link to enhancing prey capture efficiency from the results of this study.

Standing on this, I cannot agree with the authors’ another conclusion about defensive function. There are various hypothesis for the function of silk decoration, the lack of supporting evidence for prey capture/detection hypothesis does not always indicate the support for defence hypothesis.

I recommend that the authors conduct additional analysis or that the manuscript be reorganized so that the authors only state conclusions based on their findings.

Reply:

We thank the reviewer for this valuable input. In response, we have added simulations for two additional wave propagation modes and revised the conclusions accordingly. Specifically, we now include simulations of waves propagating tangentially and normally with respect to the spiral threads.

I have other concerns. The authors use the term “stabilimentum” throughout the manuscript. However, the term comes from “to stabilize web”. Now, a lot of other functions were proposed, as mentioned in introduction, I suppose that“silk decoration” is a more appropriate term.

Reply:

We thank the reviewer for this suggestion. While it is well known that the stabilimentum does not contribute to the mechanical stability of the web, the term stabilimentum is consistently used in the literature. For the sake of clarity and consistency with established terminology, we have chosen to retain its use. To address potential confusion, we have added/modified the following lines:

Lines 20:

“The stabilimentum, or structural decoration, in spider orb webs is a fascinating structure.”

Lines 46-47:

“A particularly intriguing aspect of these webs is the stabilimentum (plural: stabilimenta), a fascinating structural decoration of the web.”

Lines 55-57:

“Although the word suggests otherwise, it is well established that stabilimentum does not improve the structural stability of the web23.”

In introduction, the rationale of the connection between the presence of silk decoration and vibration propagation was not fully developed. No previous studies about silk decoration is directly connected with vibration. Therefore, I recommend the authors to explain more throughly why the effect of silk decoration on vibration propagation should be examined and what outcomes would be expected in introduction.

Reply:

We thank the reviewer for this suggestion. To improve the manuscript, we have added/modified the following lines in the manuscript:

Lines 67-72:

“The presence of additional structural elements in a spider web is likely to influence its dynamic properties by modifying the web’s geometry and, consequently, altering the pathways through which specific vibrations travel38. In particular, the propagation of elastic waves depends not only on the material properties but also on the geometry of the structural components through which they move. Therefore, in principle, the propagation of elastic waves in webs containing a stabilimentum should differ from that in webs without such a structure.”

L77. I happened to be aware that the phrase “is added to neighboring radii in a zigzag manner” coincides perfectly with one appeared in Foelix’s book. Although the book is referred, I recommend the authors to rephrase it.

Reply:

We thank the reviewer for pointing this out. Indeed, we based our description of how spiders add the stabilimentum on Foelix’s book. We have rephrased the sentence accordingly.

Figure 1c. The scheme for Juvenile type is somewhat different from actual silk decoration appeared in the photo.

Reply:

We thank the reviewer for noticing this. We have modified the figure accordingly.

Reviewer 2:

I am a biologist, and have no expertise in bio-materials, and I thus cannot evaluate the major aspects of this paper relating to quantification of elastic wave propagation and the resulting perspectives on bio-inspired metamaterials. I have only two comments in this area. I wonder why the authors used only out-of-plane vibrations, when previous work (Klärner, D., and F. G. Barth. 1982. Vibratory signals and prey capture in orb-weaving spiders (Zygiella x-notata, Nephila clavipes; Araneidae). Journal of Comparative Physiology A: Neuroethology, Sensory, Neural, and Behavioral Physiology 148:445–455) concluded that, for the spider, the most important vibrations of non-sticky lines in orbs are longitudinal rather than tranverse. This gives reason to question the biological implications of this work on elastic wave propagation, but does not necessarily impact implications regarding bio-inspired metamaterials. A second comment is that the spider’s own weight, while she rests at the hub, must alter the tensions on different radii and hub lines substantially. Especially strong increases in tensions would be expected at the two sites held by the tarsi of the spider’s legs IV (increasing tensions above and decreasing just below these tarsi); such tension modifications would be expected to affect transmission of vibrations. I do not see how this effect was included in the simulations.

Reply:

We thank the reviewer for raising these two important points. We acknowledge that our initial study was limited in the types of waves analyzed. To address this, we have added simulations of two additional wave types: waves propagating tangentially to the spiral threads and normally (the new data and text were not highlighted for the sake of readability).

We also agree that the spider’s weight is an important factor in this analysis. In our study, we considered a spider weighing 20 mg, with its weight evenly distributed across the radial threads. While the magnitude of the weight itself is not expected to significantly affect the study’s outcomes, we recognize that different weight distributions could have an impact. We have acknowledged this limitation in the manuscript and are currently investigating it for a future study, as this will require an extensive analysis of spider positioning on the webs and the development of a new set of simulations.

We invite the reviewer to examine the new results regarding the additional wave types. For clarity and completeness, we have also added/modified the following lines concerning the issue of weight distribution:

Lines 318-320:

“Moreover, the spider's position on the web could play a role in mitigating/enhancing such change in prey perception time, since different positions alter spider weight distribution on the web48.”

On the biological side of the paper, I believe that the findings in this study are of doubtful importance for discussion of the possible functions of stabilimenta for the spiders. I would advise the authors to omit most of what they say about the functions of stabilimenta and focus their paper on elastic wave propagation. This would require substantial, but only a relatively simple modifications of the ms.

Reply:

We thank the reviewer for this comment. We have revised the manuscript to omit most of the biological speculations that are not supported by our data. Along with the additional analyses, we believe the manuscript now approaches the stabilimentum from a more engineering-focused perspective.

My doubts regarding the function(s) of stabilimenta stem from the following points. 1) The wide variation that the authors document in stabilimentum design (long known in this and other species) is itself an argument against the importance of its effects on elastic wave propagation; if more rapid propagation were important in promoting prey capture, then one would not expect webs with more elaborate stabilimenta (and vice versa) (the variation in stabilimentum design is explained by alternative hypotheses for stabilimentum function such as protection from visually orienting predators).

Reply:

We thank the reviewer for raising this very important point. For the sake of clarity and completeness, we have added/modified the following lines:

Lines 330-338:

“Our results contribute to a better understanding of the role stabilimenta play in shaping the dynamic properties of orb webs, revealing that prey detection seems to be influenced only by vibrations propagating tangentially to the spiral threads. However, from a biological standpoint, the high variability in stabilimentum geometry suggests that the observed differences in elastic wave propagation are unlikely to have a consistent or significant functional role. For this reason, our findings are better interpreted from an engineering perspective. They open up new possibilities for the design of bio-inspired metamaterials with tunable anisotropic wave propagation properties—for example, spiderweb-inspired lattices that allow controlled delays in wave transmission along specific pathways.”

2) The fact that many orb weaving species remove the entire central portion of the hub would require additional explanation if propagation of elastic waves in the hub has an important effect promoting prey capture.

Reply:

We agree with the reviewer on this point. However, in the present work, we focus on a simplified model of the web of Argiope bruennichi, as a detailed comparative study across different species would require further extensive investigation.

For the sake of completeness and clarity, we have added/modified the following lines:

Lines 344-346:

“Based on these observations, we performed numerical simulations on simplified web geometries to investigate how these decorations influence vibration propagation within the orb web.”

Lines 115-117:

“For the sake of simplicity, our models did not include the web hub—the central region of the web, which is often removed after construction by certain species, such as Argiope bruennichi.”

3) The first vibrations conveying the information needed by the spider at the hub regarding the location where the prey is located do not arrive from the central portions of the hub, but from beyond the hub. Propagation of elastic waves through the hub, which in Figs. 2a and 2c,d seem to suggest was the phenomenon that was modeled in this study, is not obviously related to this.

Reply:

See previous comment.

4) The constrained positions of the spider’s legs, whose mimicry of the form of the stabilimentum in many other species (including other Argiope species) is explained by alternative hypotheses of protection from predators and prey capture, but is not explained by the elastic wave propagation hypothesis.

Reply:

We thank the reviewer for this comment. In this work, we limit our analysis to the geometries of the stabilimentum and do not consider the position of the spider’s legs. We acknowledge this as a limitation and have noted it in lines 318-324. However, a detailed investigation of leg positioning is beyond the scope of the present study.

Minor points:

As the authors say, their conclusion from their survey that stabilimentum forms vary intra-specifically is not new. As such, it is data perhaps appropiate for a specialist spider journal, not PLoS. Perhaps they can document contrasts between their observations and previous studies.

Reply:

We thank the reviewer for this comment. While we understand that these results may also appeal to a more specialized audience in arachnology, we believe that the engineering perspective presented broadens their relevance. Therefore, we are confident that PLOS is an ideal platform to reach a wide and interdisciplinary audience who will benefit from these findings.

Line 36 The claim that orbs “optimize mechanical performance” in biological terms (costs of materials, construction behavior and maintenance versus payoff in materials and energy from prey captured due to vibration transmission, interception and retention of prey in nature) is not documented in the references cited (nor in any analysis of any orb web, despite frequent claims to the contrary in the literature). Various factors have been studied separately, as mentioned in the following lines in the ms; but the balance is unknown (and extremely difficult to document convincingly). The claim for optimization needs to be removed.

Reply:

We thank the reviewer for this comment. We have removed the claim.

---

## [Decision Letter · Decision Letter 1]

13 Aug 2025

PONE-D-24-51796R1The effect of different structural decoration geometries on vibration propagation in spider orb websPLOS ONE

Dear Dr. Greco,

Thank you for submitting your manuscript to PLOS ONE. After careful consideration, we feel that it has merit but does not fully meet PLOS ONE’s publication criteria as it currently stands. Therefore, we invite you to submit a revised version of the manuscript that addresses the points raised during the review process.

I thank the authors for the substantial revision that they carried out. Please refer to specific suggestions that the reviewers made to improve the manuscript and to correct any potential errors.

We look forward to receiving your revised manuscript.

Kind regards,

Shoko Sugasawa

Academic Editor

PLOS ONE

Journal Requirements:

Reviewers' comments:

Reviewer's Responses to Questions

**Comments to the Author**

1. If the authors have adequately addressed your comments raised in a previous round of review and you feel that this manuscript is now acceptable for publication, you may indicate that here to bypass the “Comments to the Author” section, enter your conflict of interest statement in the “Confidential to Editor” section, and submit your "Accept" recommendation.

Reviewer #1: (No Response)

Reviewer #3: (No Response)

2. Is the manuscript technically sound, and do the data support the conclusions?

Reviewer #1: Yes

Reviewer #3: Yes

3. Has the statistical analysis been performed appropriately and rigorously? 

Reviewer #1: N/A

Reviewer #3: Yes

4. Have the authors made all data underlying the findings in their manuscript fully available?

Reviewer #1: Yes

Reviewer #3: Yes

5. Is the manuscript presented in an intelligible fashion and written in standard English?

Reviewer #1: Yes

Reviewer #3: Yes

6. Review Comments to the Author

Reviewer #1: In the revised manuscript, the authors had incorporated the two other types of vibration, in addition to transverse vibration, in investigating the effect of stabilimentum on the transmission of vibratory signals through spider’s orb web. I feel that the current research methodology is more consistent with the objectives of this study than the original manuscript. Nevertheless, several minor concerns remain in the organization of the manuscript, especially for the presentation of the results and significance of this study. I have listed them below.

—

I suggest that the authors might add some sentences to explain the importance of vibration transmissiion through the web to spider’s foraging success in introduction, perhaps before explaining the effect of stabilimentum on vibration transmission, i.e., line 67. I saw such an explanation in lines 305-306, but it had better be appeared earlier. Here, the author may better introduce that some spiders are considered to control how vibration signal emitted from trapped prey travel through the web via adjusting thread tension by adding stabilimentum (Watanabe 1999) or behaviorally (Nakata 2010, 2013)

Nakata, K. (2010). Attention focusing in a sit-and-wait forager: a spider controls its prey-detection ability in different web sectors by adjusting thread tension. Proceedings of the Royal Society B: Biological Sciences, 277, 29-33. https://doi.org/10.1098/rspb.2009.1583

Nakata, K. (2013). Spatial learning affects thread tension control in orb-web spiders. Biology Letters, 9(4). https://doi.org/10.1098/rsbl.2013.0052

Lines 91-92: In Figure 1a, the spider seems to wrap a prey with silk, not spin stabilimentum. Although it’s true that spiders use the same type or silk for wrapping prey and stabilimentum, I am afraid that the text here is misleading. Additionally, the authors may refer a recent literature by Lee and Moon (2025).

Lee, S. M., & Moon, M. J. (2025). Structural and Functional Analyses of Stabilimentum in the Garden Spider, Argiope bruennichi (Araneae: Araneidae). Entomological Research, 55(2), e70023.

Lines 122-126: In the literature studying vibration transmission through webs, these three types of vibrations are often referred to as longitudinal, transverse, and lateral vibrations. For consistency, I suggest that authors follow this nomenclature or at least explain the relationship between the terms used in this study and those used in the previous literatures.

Lines 202-203: In Figure 3, the lines for each type of stabilimentum could not be seen, except for line A. Did this happen because all the lines appear to overlap with little difference between them? If so, it should be better explained in Figure legend. I had similar quesions and comments for Figure 4 and 5.

Lines 228-239: It was not easy for me to understand that whether stabilimentum decreased time necessary to achieve threshold displacement. Line 228 read there was a mild effect, but line 237 read there was no significant effect. I understood these sentences were not contradictory with each other only after reading lines 242-243. If the authors considered that the decrease in time caused by N stabilimentum when the prey location is orthogonal to the direction of stabilimentum was exceptional case, consider the order of presenting results accordingly.

Reviewer #3: The paper is interesting and topical. The idea is original and the analyses seem to be well implemented.

I can see the paper has had a review already so I see that the previous reviewers provided detailed feedback that the author has now implemented.

I do want to pick up on some issues and/or erroneous pieces of information presented. I expect the authors to address these in order to complete this nice work.

Stabilimentum are sometimes just called web decorations, as per the explanations in Herberstein et al. 2000. Please explain this as it helps readers find relevant literature.

Stabilimentum does not need to be italicized.

Much of the literature presented herein is somewhat outdated. I recommend the authors refer to the paper by Yeh et al. 2015 in Scientific Reports on the simultaneous top down (from predators) and bottom up (from prey) pressures acting on stabilimentum form selection by individuals. The important point made here is about prey (primarily honeybees) learning to distinguish certain stabilimentum designs and avoiding webs, and predators (primarily wasps) learning to recognize and use them to hunt spiders.

Please add some lines about this work (as well as that by Cheng et al. 2010 in J. Exp Biol. on insect form vision implications) within the Introduction. It will also assist you to more strongly argue a case for their being large natural variations in stabilmenta use, to which you can argu there is a neglected cost – that of reduced vibratory propagation (thus reduced mating and feeding opportunities).

Line 62-63: Stabilimenta absolutely do emit a stronger UV signal than other spider silk. See Blamires et al. 2019 (J Roy Soc Interface) and 2020 (Roy Soc Open Sci) for spectra measured on Major Ampullate silks for instance. The spectra show a deep depression in the UV range.

Line 65: protection against predators is certainly not the most well supported hypothesis for stabilimentum use by Argiope spp. (it is for Cyclosa spp. nonetheless). Prey luring is by far more widely shown. Again refer to Yeh et al. and the explanation therein as to how the 2 forces shape stabilimentum use.

Give more detail in your methods about how you controlled variables and ran the simulations as well as the actual web stabilimenta variations considered.

There are form variations not represented in Figure 1. Explain these and how including them in a model might affect the outcomes.

7. PLOS authors have the option to publish the peer review history of their article (what does this mean? ). If published, this will include your full peer review and any attached files.

**Do you want your identity to be public for this peer review?** For information about this choice, including consent withdrawal, please see our Privacy Policy .

Reviewer #1: No

Reviewer #3: **Yes: ** Sean Blamires

---

## [Author Response · Author response to Decision Letter 2]

28 Aug 2025

Reviewer 1:

"In the revised manuscript, the authors had incorporated the two other types of vibration, in addition to transverse vibration, in investigating the effect of stabilimentum on the transmission of vibratory signals through spider’s orb web. I feel that the current research methodology is more consistent with the objectives of this study than the original manuscript. Nevertheless, several minor concerns remain in the organization of the manuscript, especially for the presentation of the results and significance of this study. I have listed them below."

Reply:

We thank the reviewer for this positive comment. We have modified the manuscript and we hope that it is now considered suitable for publication.

"I suggest that the authors might add some sentences to explain the importance of vibration transmissiion through the web to spider’s foraging success in introduction, perhaps before explaining the effect of stabilimentum on vibration transmission, i.e., line 67. I saw such an explanation in lines 305-306, but it had better be appeared earlier. Here, the author may better introduce that some spiders are considered to control how vibration signal emitted from trapped prey travel through the web via adjusting thread tension by adding stabilimentum (Watanabe 1999) or behaviorally (Nakata 2010, 2013).

Nakata, K. (2010). Attention focusing in a sit-and-wait forager: a spider controls its prey-detection ability in different web sectors by adjusting thread tension. Proceedings of the Royal Society B: Biological Sciences, 277, 29-33. https://doi.org/10.1098/rspb.2009.1583

Nakata, K. (2013). Spatial learning affects thread tension control in orb-web spiders. Biology Letters, 9(4).

Reply:

We thank the reviewer for this valuable input. We have added the suggested discussion and the suggested references.

"Lines 91-92: In Figure 1a, the spider seems to wrap a prey with silk, not spin stabilimentum. Although it’s true that spiders use the same type or silk for wrapping prey and stabilimentum, I am afraid that the text here is misleading. Additionally, the authors may refer a recent literature by Lee and Moon (2025).

Lee, S. M., & Moon, M. J. (2025). Structural and Functional Analyses of Stabilimentum in the Garden Spider, Argiope bruennichi (Araneae: Araneidae). Entomological Research, 55(2), e70023."

Reply:

We thank the reviewer for this suggestion. We have improved the text to avoid confusion and added the suggested reference.

"Lines 122-126: In the literature studying vibration transmission through webs, these three types of vibrations are often referred to as longitudinal, transverse, and lateral vibrations. For consistency, I suggest that authors follow this nomenclature or at least explain the relationship between the terms used in this study and those used in the previous literatures."

Reply:

We thank the reviewer for this suggestion. We have now explained the relationship among the terms, for the term lateral might sound odd from the engineering point of view.

"Lines 202-203: In Figure 3, the lines for each type of stabilimentum could not be seen, except for line A. Did this happen because all the lines appear to overlap with little difference between them? If so, it should be better explained in Figure legend. I had similar quesions and comments for Figure 4 and 5."

Reply:

We thank the reviewer for pointing this out. Indeed, the lines overlap. For this reason, we have modified the text accordingly.

"Lines 228-239: It was not easy for me to understand that whether stabilimentum decreased time necessary to achieve threshold displacement. Line 228 read there was a mild effect, but line 237 read there was no significant effect. I understood these sentences were not contradictory with each other only after reading lines 242-243. If the authors considered that the decrease in time caused by N stabilimentum when the prey location is orthogonal to the direction of stabilimentum was exceptional case, consider the order of presenting results accordingly."

Reply:

We thank the reviewer for this comment. We have now modified the text to make it clearer, and referred to the specific cases (e.g., N or A types of stabilimentum).

Reviewer 3:

"The paper is interesting and topical. The idea is original and the analyses seem to be well implemented.

I can see the paper has had a review already so I see that the previous reviewers provided detailed feedback that the author has now implemented.

I do want to pick up on some issues and/or erroneous pieces of information presented. I expect the authors to address these in order to complete this nice work."

Reply:

We thank the reviewer for his useful comments. We have improved the manuscript accordingly.

"Stabilimentum are sometimes just called web decorations, as per the explanations in Herberstein et al. 2000. Please explain this as it helps readers find relevant literature.

Stabilimentum does not need to be italicized."

Reply:

We thank the reviewer for this comment. We have revised the manuscript to explain the terminology and remove italic font for the word stabilimentum.

"Much of the literature presented herein is somewhat outdated. I recommend the authors refer to the paper by Yeh et al. 2015 in Scientific Reports on the simultaneous top down (from predators) and bottom up (from prey) pressures acting on stabilimentum form selection by individuals. The important point made here is about prey (primarily honeybees) learning to distinguish certain stabilimentum designs and avoiding webs, and predators (primarily wasps) learning to recognize and use them to hunt spiders. "

Reply:

We thank the reviewer for this input. We have now added the suggested reference alongside the ones suggested by reviewer 1.

"Please add some lines about this work (as well as that by Cheng et al. 2010 in J. Exp Biol. on insect form vision implications) within the Introduction. It will also assist you to more strongly argue a case for their being large natural variations in stabilmenta use, to which you can argu there is a neglected cost – that of reduced vibratory propagation (thus reduced mating and feeding opportunities). "

Reply:

We thank the reviewer for the suggestion. We have inserted a sentence related to the paper in the introduction.

"Line 62-63: Stabilimenta absolutely do emit a stronger UV signal than other spider silk. See Blamires et al. 2019 (J Roy Soc Interface) and 2020 (Roy Soc Open Sci) for spectra measured on Major Ampullate silks for instance. The spectra show a deep depression in the UV range."

Reply:

We thank the reviewer for suggesting these interesting works, which have now been implemented in the Introduction.

"Line 65: protection against predators is certainly not the most well supported hypothesis for stabilimentum use by Argiope spp. (it is for Cyclosa spp. nonetheless). Prey luring is by far more widely shown. Again refer to Yeh et al. and the explanation therein as to how the 2 forces shape stabilimentum use."

Reply:

We thank the reviewer for this comment. The reference has been added following the previous comments, and here we have now modified the text to be less conclusive in our statement.

"Give more detail in your methods about how you controlled variables and ran the simulations as well as the actual web stabilimenta variations considered.

There are form variations not represented in Figure 1. Explain these and how including them in a model might affect the outcomes."

Reply:

We thank the reviewer for these comments. We have added two sentences in the Methods section to provide additional details on the simulations and on how different geometries may influence them. We would like to emphasize that these simulations should be interpreted qualitatively rather than quantitatively, as several parameters can significantly affect the outcomes. In the Supplementary Information, we have already highlighted some of the most relevant factors, such as prey body mass and the distribution of the spider’s weight on the radial threads.

---

## [Editor Report · Decision Letter 2]

3 Sep 2025

The effect of different structural decoration geometries on vibration propagation in spider orb webs

PONE-D-24-51796R2

Dear Dr. Greco,

We’re pleased to inform you that your manuscript has been judged scientifically suitable for publication and will be formally accepted for publication once it meets all outstanding technical requirements.

Kind regards,

Shoko Sugasawa

Academic Editor

PLOS ONE

---

## [Editor Report · Acceptance letter]

PONE-D-24-51796R2

PLOS ONE

Dear Dr. Greco,

I'm pleased to inform you that your manuscript has been deemed suitable for publication in PLOS ONE. Congratulations! Your manuscript is now being handed over to our production team.

Kind regards,

on behalf of

Dr. Shoko Sugasawa

Academic Editor

PLOS ONE